# Digital Twin Smart City Visualization with MoE-Based Personal Thermal Comfort Analysis

**DOI:** 10.3390/s25030705

**Published:** 2025-01-24

**Authors:** Hoang-Khanh Lam, Phuoc-Dat Lam, Soo-Yol Ok, Suk-Hwan Lee

**Affiliations:** Department of Computer Engineering, Dong-A University, Busan 49315, Republic of Korea; 2371049@donga.ac.kr (H.-K.L.); 2478532@donga.ac.kr (P.-D.L.)

**Keywords:** digital twin, smart city, deep learning, thermal comfort, Cesium, Unreal Engine

## Abstract

Digital twin technology us used to create accurate virtual representations of objects or systems. Digital twins span the object’s life cycle and keep updated with real-time data. Therefore, their simulation capabilities can be combined with deep learning to create a system that simulates scenarios, enabling analysis. As cities continue to grow and the demand for sustainable development increases, digital twin technology, combined with AI-driven analysis, will play a critical role in shaping the future of urban environments. The ability to accurately simulate and manage complex systems in real time opens up new possibilities for optimizing energy usage, reducing costs, and improving the quality of life for urban residents. In this study, a digital twin application is built to visualize a smart area in South Korea, utilizing a deep learning model for personal thermal comfort analysis, which can be useful for managing and saving building and household energy consumption. Using Cesium for Unreal, a powerful tool for integrating 3D geospatial data, and leveraging DataSmith to convert 3D data into Unreal Engine format, this study also contributes a roadmap for smart city application development, which is currently considered to be lacking. By creating a robust framework for smart city applications, this research not only addresses current challenges but also lays the groundwork for future innovations in urban planning and management.

## 1. Introduction

A city is a combination of many components, geometric objects (buildings, houses, and streets), semantics (security and network traffic), and sensors (temperature, humidity, and air quality sensors). With the rapid development of cities, monitoring and managing these components is essential for improving quality of life and saving energy. Smart cities (SCs) have been defined by International Business Machines Corporation (IBM) as cities that use information and communication technology (ICT) to sense, analyze, and integrate the key information of their core systems [1]. In recent years, many studies have shown interest in 3D city modeling and digital twins, as they allow for the creation smart city applications in which decision makers can perform analyses, simulations, planning, and monitoring in different domains (energy management, urban planning, and weather simulation). Many approaches have been proposed, such as CityGML, CityJson, and the combination of Building Information Modeling (BIM) and Geographic Information System (GIS) modeling to provide City Information Modeling (CIM) [2]. Lam et al. [3] successfully implemented a method for converting Industry Foundation Class (IFC) data to 3D city models and evaluated the results through visualization across multiple applications. This method lays a promising foundation for future smart city applications, as it can comprehensively map all IFC data to 3D city models, enabling the seamless integration of building information modeling (BIM) into smart city platforms. As shown in Figure 1, a smart city includes four main components: natural components (e.g., world terrain and Sun), geometric objects (static and mobile objects), semantics (e.g., noise level, network traffic, etc.), and sensors (e.g., weather sensors, etc.). These are the basic elements required to monitor and manage a smart city. Cities: Skyline, a popular game, can be described as a single-player, open-ended city-building simulation [4]. Players engage in urban planning by establishing a road network; controlling zoning; and providing public services such as budgeting, education, employment, pollution management, etc. Based on this concept, leveraging a 3D city model, we focus on visualizing housing objectsand weather conditions to keep track of rapidly changing conditions.

A contour map was built by Hao Wang et al. [5], with colors from cool to warm to reflecting low to high research intensity by topic, describing the distribution and concentration of studies based on the time scale and knowledge management. Figure 2 highlights a red area for transportation/mobility in 2020, indicating that this sector is the most extensively researched. Additionally, warm-toned areas are visible in other aspects of smart cities (SCs), such as energy/power systems, environmental and resource management, and public infrastructure management, suggesting frequent applications of digital twin (DT) technology in these domains. The figure also reveals that DTs have not yet been widely adopted in smart education (9) and the smart economy (10). Overall, research on DT technology has garnered significant attention in recent years. An early fire detection system was trained using synthetic data, following the method proposed in [6], demonstrating its potential as a use case within DT environments. Geographic Virtual City Miniature (GeoVCM) [7] demonstrates an application that integrates both real and virtual IoT devices into a digital twin (DT) platform, allowing city managers to interact with and monitor urban environments in real time. A DT version of a two-story building in Larisa, northern Greece, was developed in [8], allowing managers to monitor sensor values in each room and track the building’s electricity consumption. The Intelligent Transportation System [9] incorporates numerous IoT devices to integrate multi-dimensional and multi-state data into a collaborative data framework. This system enables the visualization of traffic flow and other details related to vehicles on the implemented roads, providing real-time insights into traffic conditions and vehicle movements.

The indoor environment is the main factor affecting the health and well-being of residents/occupants. Thermal comfort refers to the condition of being physically and psychologically comfortable with the surrounding environment, especially the temperature. It is measured by various indices, such as Predicted Mean Vote (PMV), Predicted Percentage of Dissatisfied (PPD), and thermal sensation vote, which are the most prominent conventional methods [10]. Thermal comfort (TC) has been applied in buildings and houses by designing, suggesting, and controlling heating, ventilation, and air conditioning (HVAC) systems to maintain an optimal indoor temperature, relative humidity, and air movements. HVAC systems consume up to 50% of the total energy budget of a building, amounting to a staggering 20% of the total energy consumption in the USA [11]. There is a worldwide standard, ASHRAE 55 [12], determining a satisfactory thermal environment in occupied spaces and compliance documents. These indices assess thermal comfort levels by incorporating factors such as temperature, humidity, air velocity, clothing insulation, and metabolic rate. The development of personal comfort models requires the collection of extensive sensor-related measurements and user-labeled data, a process that can be highly intrusive and labor-intensive. To address this, Zeynep et al. [13] proposed a hybrid active learning framework aimed at reducing data collection costs while developing data-efficient and robust models. Many researchers have referenced their study on the three popular thermal comfort metrics shown in Figure 3: Thermal Sensation Vote (TSV), Thermal Preference Vote (TPC), and Thermal Comfort Vote (TCV). A deep neural network called DeepComfort [14] was proposed based on multi-task learning to predict the three most important thermal comfort metric, achieving 90% accuracy on a combination of the ASHRAE II dataset [15] and self-collected primary student dataset. We primarily rely on TSV (thermal sensation vote) to evaluate our thermal comfort model, as this metric reflects how individuals perceive the current environmental conditions, aligning closely with our objective. Through the use of thermal comfort, we can analyze the effectiveness of HVAC systems, as well as make some suggestions to control the system to save on energy consumption. We build thermal comfort using a Mixture-of-Expert (MoE) model architecture with two main backbones, namely Long Short-Term Memory (LSTM) and a Liquid Neural Network (LNN), integrated with the transformer architecture for evaluation, analysis, and comparison.

In this study, the smart city platform successfully maintained a high frame rate, ensuring smooth exploration of the virtual environment, including villages, houses, and building structures. Each component within the platform was embedded with detailed, queryable information, enhancing the user’s ability to navigate and interact with the virtual space. The integration of the deep learning model into the platform allowed for real-time analysis of thermal comfort levels, considering individual characteristics such as height, weight, and clothing. This real-time prediction capability provided insights into each person’s comfort based on the environmental conditions. The model demonstrated robust performance, delivering accurate thermal comfort predictions across various environmental scenarios. Additionally, the system effectively adjusted to diverse body conditions, providing highly personalized comfort assessments. The combination of the smart city platform and the deep learning model enabled a seamless integration of environmental modeling and real-time comfort analysis, offering significant potential for urban planning and smart city management by optimizing both infrastructure and individual well-being.

The remainder of this paper is structured as follows. Section 2 presents the technical details of the smart city platform and the personal thermal comfort model. This section covers how environmental measurements are visualized, the features required to develop the personal thermal comfort model, and the structural design of the model. Section 3 discusses the performance of both the thermal comfort model and the smart city platform, highlighting key evaluation metrics. Finally, the conclusions of the study and potential future work are outlined in Section 4.

## 2. Materials and Methods

As shown in Figure 4, our method integrates natural components, static geographic objects, and environmental sensor data (Section 2.1.1) into a platform built using Unreal Engine, as detailed in Section 2.2. OpenWeather is utilized to collect and transfer weather-related data to both the platform and the thermal comfort model. This model (Section 2.3) is powered by personal and environmental data (Section 2.1.2) and is connected to the platform through FastAPI (Section 2.4).

### 2.1. Dataset Overview

#### 2.1.1. Platform Data

The proposed approach can be conceptualized as a three-dimensional map incorporating detailed information and interactive elements, enabling users to interact with the environment and retrieve pertinent data in real time. A key component of such a map is the globe, which features 3D terrain, along with a dynamic solar illumination system that simulates daylight conditions. Cesium is a globally recognized platform for supporting 3D geospatial applications that has been widely utilized in numerous studies ([3,16,17,18]). With the assistance of Cesium, a geospatial system is implemented, enabling accurate visualization and management of spatial data in a 3D environment. Cesium facilitates the integration of geographic information, such as building locations and terrain data, into the smart city platform, allowing for real-time mapping and analysis of urban infrastructure within a global coordinate system, as shown in Figure 5. SunSky [19] is also integrated into the system to provide realistic daylight simulation based on the date, time, and time zone of the specific area. This feature enhances the accuracy of the environmental visualization by adjusting lighting conditions dynamically according to the geographical location and time of day, offering a more immersive and contextually accurate representation of the urban environment.

Among the important components that create smart city applications are houses and buildings. To import them into the application, we converted data from Revit (.rvt) to static mesh by leveraging the DataSmith plugin. As shown in Figure 6, Datasmith is a suite of tools and plugins designed to import complete pre-constructed scenes and complex assets from various industry-standard design applications into Unreal Engine. During the conversion of a Revit file to Datasmith format, the geographic data are preserved to position the houses accurately in the correct coordinates of the Cesium globe.

The detailed information for each component is gathered, analyzed, and formatted in CityGML, which is then converted to 3D tiles for querying and visualization purposes. The conversion process begins with building information modeling (BIM) data in IFC format, which serve as the foundational input. These data are transformed into CityGML format at Level of Detail 4 (LOD4), capturing intricate architectural features. Subsequently, the CityGML data are converted into 3D Tiles format, optimized for efficient streaming and visualization. This streamlined process ensures a smooth transition from BIM to an interoperable 3D format, facilitating dynamic data utilization and real-time visualization. Cesium 3D Tiles is a geospatial data format and rendering engine crucial for visualizing and streaming 3D geospatial content over the web. It offers an efficient and scalable solution for delivering large-scale 3D geographic information, enabling users to interactively explore complex 3D environments in web-based applications and virtual globes. Cesium 3D Tiles excels in streaming and rendering detailed 3D models in real time, optimizing performance and allowing for seamless visualization of intricate data, even in large-scale or high-density scenarios.

#### 2.1.2. Thermal Comfort Data

Most deep learning-based thermal comfort (DLTC) studies utilize feature parameters that encompass a combination of indoor environmental measurements (e.g., indoor temperature), outdoor environmental data (e.g., daily rainfall), and individual-specific characteristics (e.g., clothing) [20]. Accordingly, our thermal comfort model incorporates input features consisting of indoor environmental measurements (e.g., indoor temperature and relative humidity), individual-specific factors (e.g., clothing value), and weather data obtained from OpenWeather for the corresponding month during which the field experiments were conducted. The complete list of features in the dataset is presented in Table 1.

There have been multiple attempts to develop a unified and widely accepted thermal comfort model that can be received and adopted by large audiences. The most popular model is the Predicted Mean Vote (PMV) model, which was constructed by P. O. Fanger [22] and was later adapted to American Society of Heating, Refrigerating, and Air-Conditioning Engineers (ASHRAE) Standard 55 [12]—Thermal Environmental Conditions for Human Occupancy. Many recent works have adapted PMV calculation as a useful feature, showing promising results [23,24]. The PMV value can be directly calculated using a system of highly nonlinear and iterative equations, which were eventually adopted in ASHRAE Standard 55 [12]:(1)$$PMV=(0.028+0.3033e^{-0.036M})\times L$$(2)$$\begin{array}{c} L=\left(M-V\right)-3.05\times 10^{-3}\left(5733-6.99\left(M-V\right)-Pa\right)\\ -0.42\left(M-V-58.15\right)-1.7\times 10^{-5}\left(5867-Pa\right)\\ -0.0014M\left(34-t_{a}\right)-f_{cl}h_{c}\left(t_{cl}-t_{a}\right)-3.96\\ \times 10^{-8}f_{cl}\left[{\left(t_{cl}+237\right)}^{4}-{\left(t_{r}+273\right)}^{4}\right] \end{array}$$
where *L* defines the overall heat transfer around a single occupant in W/m^2^, *M* is the metabolic rate in W/m^2^, *W* is the work emitted by the occupant in W/m^2^, Pa is the water vapor pressure, $t_{r}$ is the mean radiant temperature in °C, $t_{a}$ is the air temperature in °C, $f_{cl}$ is the clothing insulation factor defined as the percentage of the total body surface area covered by clothing, $I_{cl}$ is the clothing insulation in CLO, and $h_{c}$ is the convective-rate heat transfer coefficient.

Table 2 provides a breakdown of the dataset used in this study, categorizing samples by building type and their respective heating, cooling, and naturally ventilated conditions. This dataset provides a comprehensive basis for analyzing thermal comfort across diverse building environments.

### 2.2. Smart City Visualization Platform

Accurate visualization of urban infrastructure is crucial for informed decision making and effective city management. To support this, our objective is to develop a platform that can visualize each component of houses and buildings, along with associated data. Detailed information for house components is seamlessly transferred from Revit files into Datasmith format and stored as a UASET file, which Unreal Engine recognizes as AssetUserData for the respective objects. We implement ray tracing from the user’s camera to the mouse collision point to detect which object the user clicks on and retrieve the corresponding component’s ID. Once the component’s ID is obtained, it is used to query all relevant data, which are then displayed on a custom-designed widget on the screen. This process is illustrated in Figure 7.

Visualizing environmental measurements, such as temperature, humidity, and wind, is critical for understanding the dynamic conditions affecting urban environments. By integrating real-time or simulated data, these variables can be represented as heat maps, color gradients, or vector fields within a 3D model. For instance, temperature variations across a city can be displayed using a color-coded overlay, while wind speed and direction can be visualized using animated vectors or particle systems. Humidity levels may be represented through shading or transparency effects, providing a clear, intuitive way for users to assess the environmental conditions of buildings and infrastructure. This visualization not only enhances decision making but also aids in climate analysis and urban planning [25].

In this study, we visualize three key environmental factors: temperature, humidity, and wind. The specific techniques for visualizing each factor are described as follows:**Temperature**: Color correction with dynamic color temperature is employed to map environmental temperature values to the corresponding color. As described in Equation (Equation 3), adjusting the white balance value directly influences the temperature color, causing the visual representation of temperature to change accordingly. This allows for a more accurate depiction of environmental conditions by mapping temperature variations to specific color gradients. The color gradient ranges from orange–red (indicating hot temperatures) to blue (indicating cold temperatures), as illustrated in the legend in Figure 8.(3)W=−270×(T−40)+1500
where *W* is the white balance value and *T* is the air temperature value.**Humidity**: Since this measurement is represented as a scaled percentage value, we visualize humidity by creating a cylinder with a dynamic height corresponding to each humidity value. An example of this representation is shown in Figure 9.**Wind**: The Niagara VFX System is the primary tool for creating visual effects (VFX) in Unreal Engine 5 (UE5) [26]. By utilizing this system, we guide particle movements to follow a predefined spline object, dynamically adjusting the movement speed and angle based on the wind speed and direction. This allows for a realistic representation of environmental effects, such as wind flow, which enhances the overall visual accuracy and interactivity of the smart city platform as shown in Figure 10.(4)$$Particle_{speed}=Wind_{speed}$$(5)$$Particle_{yawDegree}=Wind_{degree}-180$$Meteorological wind direction is defined as the direction from which wind originates. For example, a northerly wind blows from the north to the south. Wind direction is measured in degrees clockwise from due north. Hence, a wind coming from the south has a wind direction of 180 degrees, while one from the east has a direction of 90 degrees.

### 2.3. Thermal Comfort Model

A mixture of experts (MoE) enables models to be trained with far less computing, which means you can dramatically scale up the model or dataset size with the same compute budget as a dense model. MoE models can help by allowing different experts to specialize in different tasks, improving performance on each. MoE models consist of two main elements:Sparse MoE layers are used instead of dense feed-forward network (FFN) layers. The experts are FFNs, which can also be more complex networks or even MoE models or hierarchical MoE models.A gate network or router determines which tokens are sent to which experts (there can be more than one expert).(6)$$y=\sum_{i=1}^{n}G{\left(x\right)}_{i}E_{i}\left(x\right)$$(7)$$G\left(x\right)=Softmax\left(x.W_{g}\right)$$

Although large batch sizes are usually better for performance, batch sizes in MoE models are effectively reduced as data flow through the active experts. For example, if a batch consists of 10 tokens, 5 tokens might end in one expert, and the other 5 tokens might end in five different experts, leading to uneven batch sizes and underutilization. Shazeer et al. [27] explored other gating mechanisms, such as noisy top-k gating. This gating approach introduces some (tunable) noise, then keeps the top k values as follows:(8)$$KeepTopK{\left(v,k\right)}_{i}=\left\{\begin{array}{c} v_{i}\;\mathrm{if}\;v_{i}\;\mathrm{in}\;\mathrm{top}\;k\;\mathrm{element}\;\mathrm{of}\;v\\ -\infty \;\mathrm{otherwise} \end{array}\right.$$(9)G(x)=Softmax(KeepTopK(x,k))

This sparsity introduces some interesting properties. Selecting a sufficiently low *k* can significantly accelerate both training and inference compared to activating multiple experts. The initial hypothesis was that routing to more than one expert is necessary for the gate to learn how to distribute tasks across different experts effectively. Therefore, at least two experts need to be selected to enable the gate to generalize its routing decisions.

We define the thermal comfort dataset in a time-series format to support future integration into real-time applications, allowing for continuous monitoring and prediction of thermal comfort levels over time. This format enables the analysis of temporal patterns and trends in environmental and individual comfort data. As a result, various well-established time-series models, such as LSTM [28] and liquid neural networks (LNNs) [29] have been considered for development. As mentioned in [29], an LNN is a specialized type of neural network that continues to learn and adapt in real time, not just during the initial training phase. LNNs are specifically designed to process and predict time-series data, making them highly suitable for dynamic environments where data patterns evolve continuously. These models are constructed using a mixture-of-experts (MoE) architecture [27], a neural network design that enhances efficiency and performance by dynamically selecting and activating a subset of specialized networks known as experts for each input. This approach allows the model to allocate computational resources more effectively, focusing on the most relevant experts for specific tasks, which leads to improved accuracy and reduced computational cost in time-series data processing and prediction.

As illustrated in Figure 11, the input data are simultaneously fed into multiple expert models and a gating model. The gating model is responsible for selecting the most appropriate expert(s) for the processing of each input based on the specific characteristics of the data. This process is managed by calculating the softmax probabilities from the gating model’s output and retaining the top *k* expert models with the highest softmax values, as proposed in [30]. This ensures that only the most relevant expert models are activated, optimizing the model’s performance and efficiency for each input.

### 2.4. Integration

The thermal comfort model is integrated with the smart city platform using FastAPI, Figure 12, a modern framework designed for building fast and efficient APIs. This connection enables real-time data exchange and interaction between the thermal comfort model and the smart city infrastructure, facilitating seamless updates and automated adjustments based on environmental conditions.

All thermally related data are stored and transferred for processing, they are fed into the thermal comfort model to generate a predicted thermal sensation vote. This predicted value is then sent to and displayed on the smart city platform, providing real-time feedback on comfort levels across the urban environment and individual features.

## 3. Results and Discussion

### 3.1. Personal Thermal Comfort Model Evaluation

To evaluate the proposed model and verify its generalization ability, K-fold cross-validation is employed with k=5. The dataset is divided into *k* equal subsets, i.e., folds. In each iteration, k−1 folds are combined to form the training set, while the remaining fold is used as the validation set. This ensures that each data sample is included in the validation set exactly once and in the training set k−1 times. After completing all iterations, the average error across all *k* folds is calculated and used to select the optimal model parameters. This approach helps mitigate the bias–variance trade-off by rotating the roles of the training and validation sets, leading to more robust and reliable model performance. The dataset was divided into three subsets for the training, validation, and testing processes using an 8:1:1 ratio. This means 80% of the data was allocated for training, 10% for validation to fine tune model parameters, and the remaining 10% for testing to evaluate the model’s performance on unseen data. This approach ensures an effective balance between model development and assessment.

In this section, we evaluate the performance of the proposed model using two key criteria: accuracy and Mean Squared Error (MSE) with mean reduction. Accuracy measures the proportion of correct predictions made by the model, while MSE quantifies the average squared difference between the predicted and actual values, as described in Equation (Equation 10).(10)$$Accuracy=\frac{1}{m}\sum_{0}^{m}\left\{\begin{array}{cc} 1, & \mathrm{if}\;abs\left(\hat{y_{i}}\right)=y_{i}\\ 0, & \mathrm{otherwise} \end{array}\right.$$(11)$$MSE=\frac{1}{m}\sum_{i=0}^{m}{\left(y_{i}-\hat{y_{i}}\right)}^{2}$$
where *m* represents the total number of samples, $\hat{y_{i}}$ denotes the predicted thermal sensation value, and $y_{i}$ is the true thermal sensation vote.

Table 3 presents the training results of mixture of experts (MoE) models for predicting personal thermal comfort using LSTM and LNN. The models were evaluated based on the number of expert models (*k*), their accuracy, and mean squared error (MSE). For LSTM models, the best performance was achieved with k=9, yielding an accuracy of 0.9403 and the lowest MSE of 0.3278. For LNN models, the optimal configuration was observed with k=13, achieving an accuracy of 0.9387 and the lowest MSE of 0.3218. These results indicate that both LSTM and LNNs perform best when using nine expert models, with LNNs slightly outperforming LSTM in terms of MSE, reflecting their ability to capture complex temporal patterns effectively. However, LSTM marginally outperforms LNNs in accuracy in this configuration.

Table 4 highlights the performance of MoE-based models using various combinations of LSTM and LNN backbones. The configurations were evaluated based on accuracy and MSE. The best-performing combination was 13 LSTM-9 LNN, achieving the highest accuracy of 0.9613 and the lowest MSE of 0.3156. This indicates that this combination strikes an optimal balance between the strengths of both LSTM and LNNs, effectively leveraging their complementary features for thermal comfort prediction. Other configurations, such as 9 LSTM-13 LNN, also showed competitive performance, with an accuracy of 0.9517 and an MSE of 0.3218, suggesting that increasing the proportion of LSTM backbones generally enhances model performance. These results demonstrate the effectiveness of combining LSTM and LNN backbones to improve prediction accuracy and reduce errors in the MoE framework. The training time for both the LSTM and LNN models increases as the number of expert models (*k*) rises, with LSTM requiring slightly less time compared to LNNs for the same *k*. For example, at k=21, the LSTM training time is 1.2 h, whereas LNNs take 1.85 h. When combining backbones, the training time is longer overall, reaching 1.93 h for the 1 LSTM-21 LNN combination. However, the combined models demonstrate a trade-off between training time and improved performance, as seen in the 13 LSTM-9 LNN model, which achieves the best accuracy (0.9613) and lowest MSE (0.3156) with a training time of 1.76 h. Thus, while training time increases with model complexity, the combined approach balances performance and time efficiency effectively.

### 3.2. Smart City Platform Performance

For interactive computer graphics applications, a suitable frame rate typically ranges from 30 to 60 FPS, which ensures smooth performance and responsiveness for general use cases like user interfaces and dashboards [31]. The hardware setup included high-performance machines equipped with NVIDIA RTX A4000 GPUs, 32GB of memory, and 500GB SSD storage to manage computationally intensive tasks such as rendering and machine learning model inference. The platform was built using Python for data processing, Unreal Engine for 3D visualization, Blueprint programming for the creation of functions, and Pytorch for deep learning model integration. Data from IoT sensors, such as temperature and humidity sensors, were transmitted via HTTP protocol, ensuring low-latency communication. Geographic data integration was achieved using CesiumJS and CityGML, allowing for precise mapping of urban infrastructure. With our proposed method, the platform maintains a frame rate of approximately 40 frames per second (fps), representing an improvement in performance for 3D smart city platforms compared to other studies [32,33,34]. This enhanced frame rate ensures smoother and more responsive visualization, contributing to a better user experience and more effective real-time data analysis. As shown in Figure 13, the platform allows users to easily navigate inside a house and query detailed information related to its components. Additionally, the material of each component is preserved and visualized, providing a clearer understanding of the house’s structure and construction, further enhancing the user’s interaction and comprehension of the built environment.

Thermal comfort in the platform is also displayed as a board for each character. This board provides the detailed information required to compute the thermal sensation vote for each character, including factors such as activity level, clothing, weight, and height. The thermal sensation vote is predicted by the thermal comfort model and transferred to the platform via FastAPI, where it is visualized on the board. This feature offers city managers valuable insights into individual thermal comfort levels, enabling more informed decisions regarding HVAC system design to optimize both comfort and energy efficiency. Some examples are illustrated in Figure 14.

## 4. Conclusions

This study presents an end-to-end approach to building a smart city platform. With the proposed method, the rendering time and frame rate are optimized, maintaining performance at a level suitable for scaling into the production stage. Additionally, detailed information about each component is transferred to the platform, offering users a more comprehensive view of the smart city. Furthermore, weather data are integrated and visualized in multiple ways, enabling city managers to monitor environmental conditions and efficiently plan city infrastructure, enhancing decision making and urban management.

A comprehensive approach is proposed to integrate a thermal comfort prediction model into a smart city platform, enabling real-time analysis and visualization of environmental factors such as temperature, humidity, and wind. By utilizing a mixture-of-experts architecture, the system dynamically selects the most relevant models for data processing, enhancing both efficiency and prediction accuracy. The thermal comfort model is built on time-series data and incorporates individual-specific features, as well as environmental measurements, making it highly adaptable for real-time applications in smart cities.

The system is integrated with the smart city platform through FastAPI, allowing for seamless data transmission, processing, and the display of thermal sensation predictions. This integration ensures that city managers have access to detailed, real-time insights into thermal comfort levels across urban areas, which can inform decision making related to energy efficiency, urban planning, and building management.

This research highlights the potential for integrating advanced machine learning models into urban management systems, offering a scalable solution for monitoring and predicting thermal comfort in dynamic environments. Future work will focus on enhancing model precision; expanding the range of environmental and individual parameters; and exploring the broader applicability of the system in areas such as energy management, climate resilience, and sustainable city development. Additionally, further exploration of hybrid models and real-time learning techniques, such as liquid neural networks, may offer avenues for improving the adaptability of the system in complex and evolving urban contexts.

## Figures and Tables

**Figure 1 sensors-25-00705-f001:**
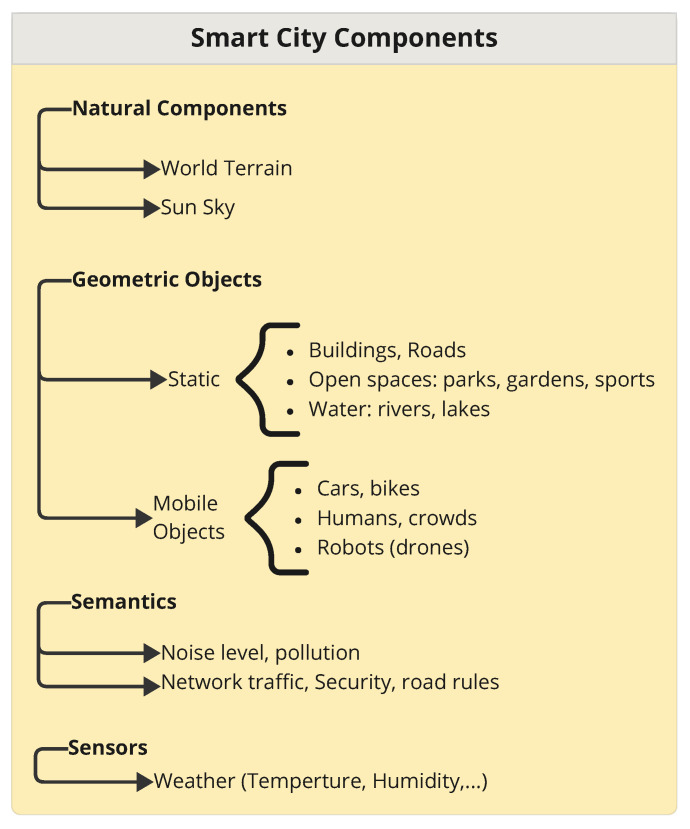
Main components of a smart city.

**Figure 2 sensors-25-00705-f002:**
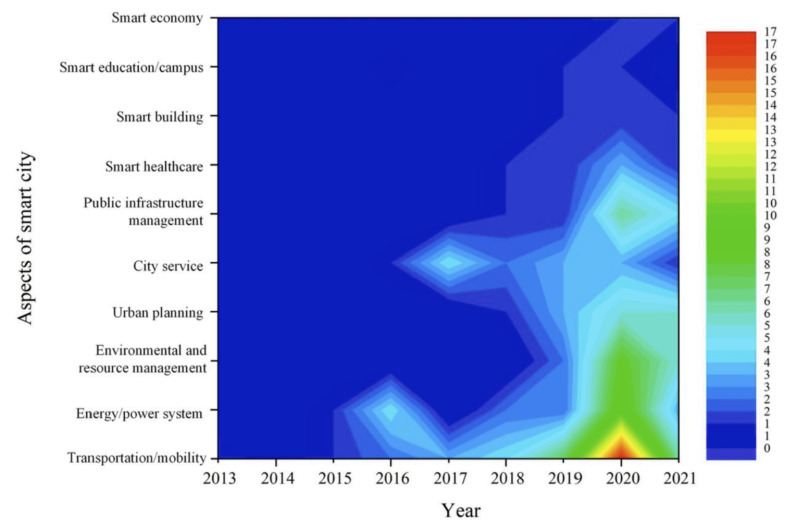
Distribution and concentration of studies on DT-supported SCs [5].

**Figure 3 sensors-25-00705-f003:**
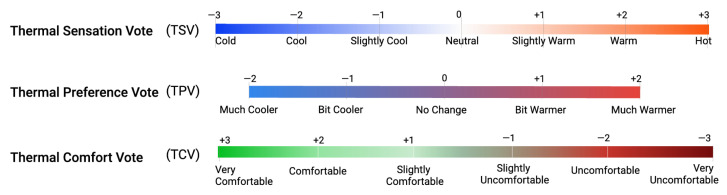
Popular thermal comfort metrics.

**Figure 4 sensors-25-00705-f004:**
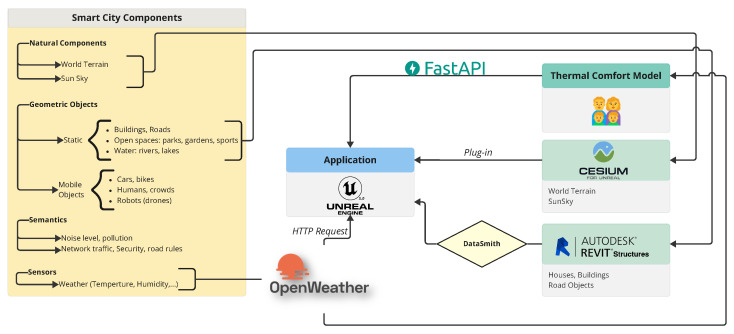
Overview of the proposed Smart City Platform.

**Figure 5 sensors-25-00705-f005:**
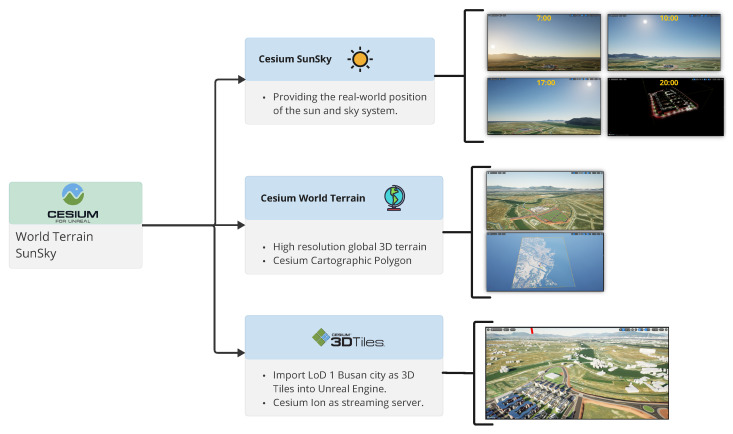
Leveraging Cesium support in Unreal Engine.

**Figure 6 sensors-25-00705-f006:**
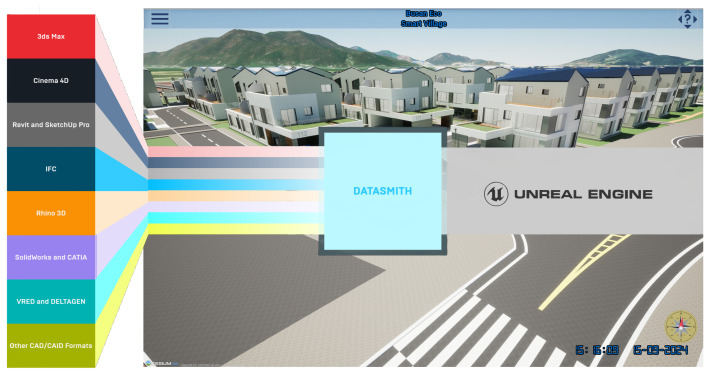
Three-dimensional houses and buildings visualized in Unreal Engine with DataSmith support.

**Figure 7 sensors-25-00705-f007:**
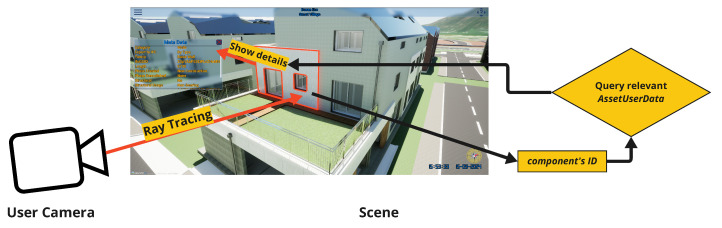
Details of house components shown by clicking (ray tracing is activated).

**Figure 8 sensors-25-00705-f008:**
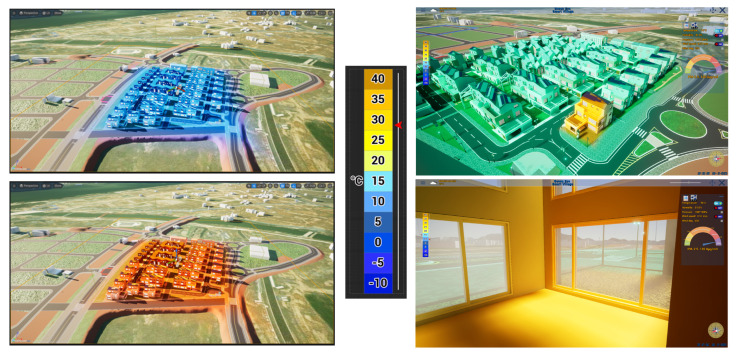
Temperature visualization in the smart city platform.

**Figure 9 sensors-25-00705-f009:**
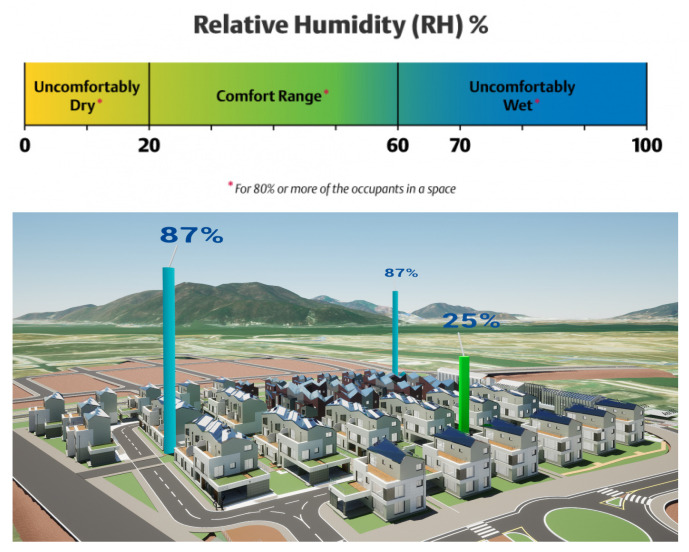
Humidity visualization in the smart city platform.

**Figure 10 sensors-25-00705-f010:**
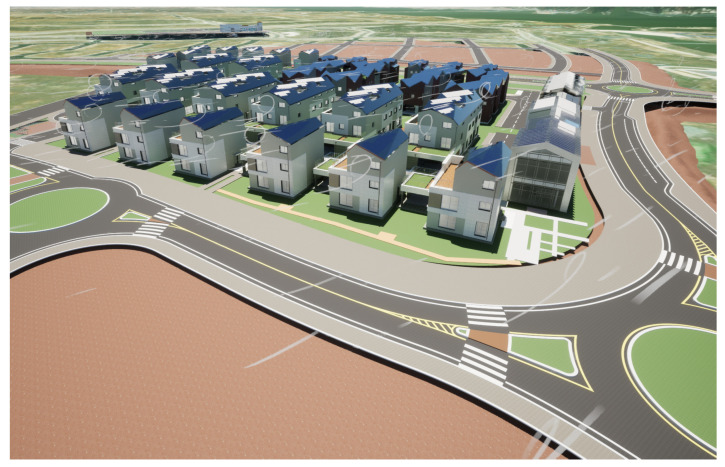
Wind visualization in the smart city platform.

**Figure 11 sensors-25-00705-f011:**
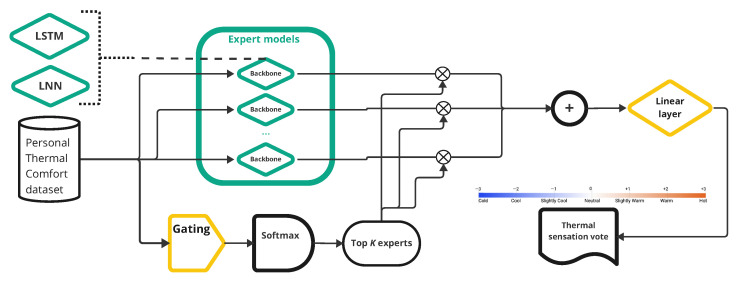
Personal thermal comfort model structure.

**Figure 12 sensors-25-00705-f012:**
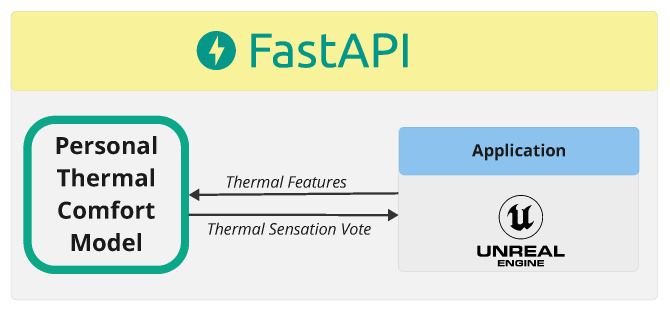
Overview of the connection between the personal thermal comfort model and the smart city platform.

**Figure 13 sensors-25-00705-f013:**
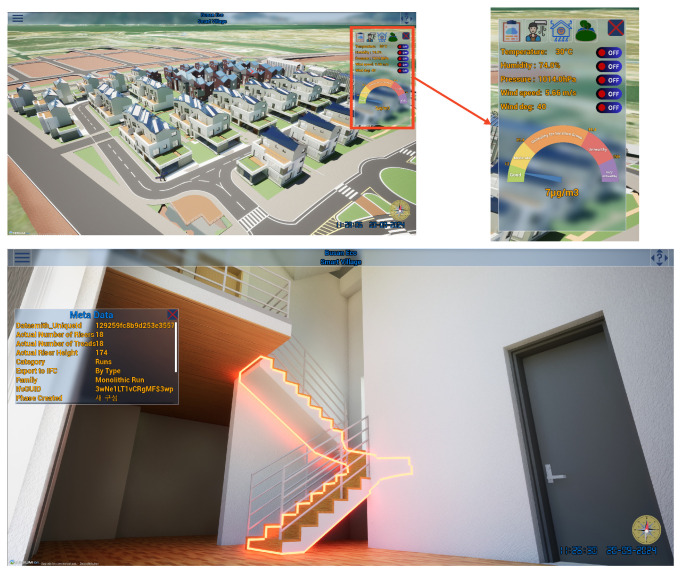
Overview of the smart city platform dashboard. Korean word in the Meta Data board means “new configuration” in English.

**Figure 14 sensors-25-00705-f014:**
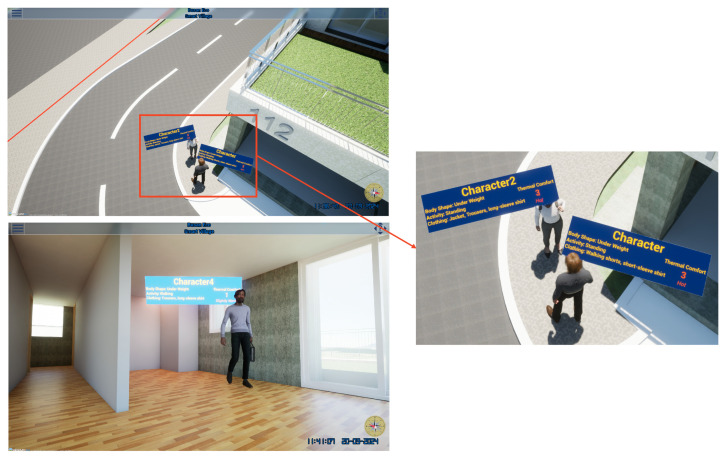
Thermal comfort information board.

**Table 1 sensors-25-00705-t001:** Coding conventions include personal details of the subjects, indoor, outdoor measurements, and data on subjective thermal comfort assessments.

Field	Description
**A. Individual-specific**	
A1. Age	Age of the subjects
A2. Height	Height of the subjects (cm)
A3. Weight	Weight of the subjects (kg)
A4. Gender	Male, female, or undefined
A5. Clothing Insulation	Intrinsic clothing ensemble insulation of the subjects (clo)
A6. Metabolic Rate	Activity status at the time of the survey (met); refer to ASHRAE 55-2020 [21]
**B. Indoor environmental measurements**	
B1. Air temperature	Air temperature measured in the occupied zone (°C)
B2. Air Velocity	Air velocity measured in the occupied zone (m/s)
B3. Relative Humidity	Relative humidity measured in the occupied zone (%)
**C. Outdoor environmental data**	
C1. Season	Summer season, winter season, or transition season
C2. Climate zone	Severe Cold Zone (SC), Cold Zone (C), Hot Summer & Cold Winter Zone (HSCW), Hot Summer & Warm Winter Zone (HSWW), Mild Zone (M)
**D. Thermal Comfort Information**	
D1. Thermal Sensation Vote	ASHRAE thermal sensation vote, from −3 (cold) to +3 (hot)
D2. PMV	Predicted mean vote, it was calculated according to ASHRAE 55-2020 [21]

**Table 2 sensors-25-00705-t002:** Number of samples in the dataset.

Building Type	Cooling	Heating	Naturally Ventilated	Total
Office	6330	6880	1735	14,945
Classroom	1202	1978	1647	4827
Residence	1010	3950	9043	14,003
Total	8542	12,808	12,425	33,775

**Table 3 sensors-25-00705-t003:** MoE-based personal thermal comfort training results. Higher is better for Accuracy while lower is better for MSE.

*k* (Number of Used Expert Models)	Training Time (h)	Accuracy ↑	MSE ↓
**LSTM**			
1	0.31	0.9400	0.3398
5	0.42	0.9370	0.3315
9	0.51	**0.9403**	**0.3278**
13	0.64	0.9201	0.3405
17	0.89	0.9145	0.3511
21	1.2	0.9044	0.3689
**LNN**			
1	0.45	0.9154	0.4652
5	0.58	0.9215	0.3513
9	1.21	0.9269	0.3249
13	1.42	**0.9387**	**0.3218**
17	1.51	0.9251	0.3409
21	1.85	0.9163	0.4783

**Table 4 sensors-25-00705-t004:** Performance of MoE-based models with different numbers of backbone combinations.

*k* (Combined Backbones)	Training Time (h)	Accuracy ↑	MSE ↓
1 LSTM-21 LNN	1.93	0.9367	0.3402
5 LSTM-17 LNN	1.88	0.9478	0.3265
9 LSTM-13 LNN	1.85	0.9517	0.3218
13 LSTM-9 LNN	1.76	**0.9613**	**0.3156**
17 LSTM-5 LNN	1.64	0.9483	0.3247
21 LSTM-1 LNN	1.52	0.9512	0.3226

## Data Availability

No new data were created or analyzed in this study. Data sharing is not applicable to this article.

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
