# Peer review of "Digital Twin Smart City Visualization with MoE-Based Personal Thermal Comfort Analysis"

_sensors, 2025, doi:10.3390/s25030705_

Round 1

Reviewer 1 Report

Comments and Suggestions for Authors

An important function of the digital twin is to assist real-time decision-making. The paper discusses a lot about how mode prediction can be improved from real data. However, I do not find discussions on the types of decisions that are made in the scenarios and how the decisions are affected by the data.

Author Response

Thank you for your valuable comments and contributions. Please refer to the attached document for our detailed responses and revisions.

Reviewer 2 Report

Comments and Suggestions for Authors

The authors present a Digital Twin platform for monitoring a smart city. The topic is really interesting but I find that there are areas in which the manuscript should be improved if it is to be published:

- Use of English should be improved, there are a lot of errors that make reading the document cumbersome.

- The authors should mention the related work in a separate paragraph and make clear how their contribution differs from others/improves challenging aspects

- The authors compare two different approaches LSTM and LNN for the development of thermal comfort module. The dataset details should be provided, meaning total size, training, validation and test sizes, time to train the models, etc. Also, the number of expert models does not seem to play a significant role since the difference is very small between k=1 and k=9 for both models.

- The authors claim that the platform achieves a frame rate of approximately 40 FPS and cite two publications to show that it is an improvement over other works. These publications are from 2018 and 2019 respectively and in my opinion quite dated for the topic of interest. Moreover, the necessary computing requirements for achieving this frame rate should be provided.

Comments on the Quality of English Language

The manuscript needs significant editing in order to correct errors in English language and improve the quality of the presentation of the relevant concepts.

Author Response

Thank you for your valuable comments and contributions to our manuscript. We appreciate your insights, and we've carefully addressed your feedback. Please refer to the attached document for our detailed responses.

Round 2

Reviewer 2 Report

Comments and Suggestions for Authors

The authors have addressed most of the comments I made in my previous review. What I believe is still missing is the time required for training the models. The quality of English has been improved.
